# Length of stay following percutaneous left atrial appendage occlusion: Data from the prospective, multicenter Amplatzer Amulet Occluder Observational Study

Kerstin Piayda[1], Shazia Afzal[1], Jens Erik Nielsen-Kudsk[2], Boris Schmidt[3], Patrizio Mazzone[4], Sergio Berti[5], Sven Fischer[6], Juha Lund[7], Matteo Montorfano[8], David Hildick-Smith[9], Ryan Gage[10], Hong Zhao[10], Tobias Zeus[1]*

1 Medical Faculty, Division of Cardiology, Pulmonology and Vascular Medicine, Heinrich-Heine-University Düsseldorf, Düsseldorf, Germany, 2 Department of Cardiology, Aarhus University Hospital, Aarhus, Denmark, 3 Cardioangiologisches Centrum Bethanien, Agaplesion Markus Krankenhaus, Medizinische Klinik 3 –Kardiologie, Frankfurt, Germany, 4 Arrhythmology and Cardiac Pacing Unit, Ospedale San Raffaele, Milan, Italy, 5 Department of Interventional and Diagnostic Cardiology, Fondazione Toscana Gabriele Monasterio, Pisa, Italy, 6 Department of Cardiology, Harzklinikum Dorothea Christiane Erxleben GmbH, Quedlinburg, Germany, 7 Heart Center, Turku University Hospital, Turku, Finland, 8 Interventional Cardiology Unit, Ospedale San Raffaele, Milan, Italy, 9 Sussex Cardiac Center, Brighton and Sussex University Hospitals, Brighton, United Kingdom, 10 Structural Heart, Abbott, St. Paul, Minnesota, United States of America

* zeus@med.uni-duesseldorf.de

**Data Availability Statement:** No data are available on a public platform. Individual deidentified participant data cannot be shared. The sponsor of

## Abstract

### Aims

To evaluate factors influencing the length of stay in patients undergoing percutaneous left atrial appendage occlusion (LAAO).

### Methods and results

Patient characteristics, procedural data and the occurrence of serious adverse events were analyzed from the Amplatzer™ Amulet™ Occluder Observational Study. Patients were divided into three groups: same day (S, 0day, n = 60, 5.6%) early (E, 1day, n = 526, 48.9%), regular (R, 2-3days, n = 338, 31.4%) and late (L, ≥4days, n = 152, 14.1%) discharge and followed up for 60 days. Procedure and device related SAE during the in-hospital stay (S: 0.0% vs. E: 1.0% vs. R: 2.1% vs. L: 23%, p<0.0001) were a major trigger for a prolonged in-hospital stay. Of the 37 subjects in the late discharge group with an SAE prior to discharge, cardiac or bleeding complications were the most common underlying conditions, occurring in 26 subjects. Multinomial logistic analysis only identified HAS-BLED score as an independent influencing factor (p = 0.04) for a late discharge. After 60 days, mortality tended to be greatest in the late discharge group (S: 0.0% vs. E: 1.0% vs. R: 1.2% vs. L: 3.3%, p = 0.1066).

this study (Abbott) is unable to legally distribute/provide access to the study data at the time of publication due to privacy concern. However, Abbott provides contact information instead where an interested researcher could apply to gain access to the data Website: https://www.cardiovascular.abbott/us/en/hcp/investigator-sponsored-studies.html Email: abbottissrequests@abbott.com.

**Funding:** This study was supported by Abbott in the form of salaries for RG and HZ and Harzklinikum Dorothea Christiane Erxleben GmbH in the form of a salary for SF. The specific roles of these authors are articulated in the 'author contributions' section. The funders had no role in study design, data collection and analysis, decision to publish, or preparation of the manuscript. No additional external funding was received for this study.

**Competing interests:** The authors have read the journal's policy and have the following competing interests: JENK has served as a proctor for Abbott and is a consultant for Abbott and Boston Scientific. PM has served as a consultant for Abbott, Boston Scientific, and Medtronic. SB has served as a proctor for Abbott. SF has served as a proctor for Biotronik and Boston Scientific and is a consultant for Abbott and employee of Harzklinikum Dorothea Christiane Erxleben GmbH. JL has served as a proctor for Abbott. MM has served as a proctor for Abbott, Boston Scientific, and Edwards Lifesciences. DHS has served as a proctor and consultant for Abbott. RG and HZ are employees of Abbott. TZ has received consulting fees, travel expenses or study honorariums from Medtronic and Edwards Lifesciences outside of this work. This does not alter our adherence to PLOS ONE policies on sharing data and materials. There are no patents, products in development or marketed products associated with this research to declare.

## Conclusion

Over half of the subjects receiving an Amplatzer Amulet occluder were discharged within 1 day of the implant procedure. Serious adverse events were a major trigger for a late discharge after LAAO. Increased HAS-BLED score was associated with a prolonged in-hospital stay.

## Introduction

Atrial fibrillation (AF) is the most frequently occurring cardiac arrhythmia worldwide and represents a major cause for morbidity and mortality in health-care systems around the world [1,2]. Population-based studies [3,4] show that the incidence and prevalence of AF are steadily increasing with a life time risk for AF of approximately 23%. Since stroke is the most feared complications of AF, potentially leading to death or serious disability, stroke prevention is a substantial part of AF patient management. Symptom relief may be offered through antiarrhythmic drugs or catheter ablation techniques, but current treatments do not reliably prevent the occurrence of thromboembolic events. According to the guidelines a long-term anticoagulative strategy remains the first line therapy for stroke prevention with a Class I Level A recommendation, despite evident side effects and limited drug adherence [5,6]. However, percutaneous left atrial appendage occlusion (LAAO) has emerged as a favorable non-pharmacological stroke prevention. After long term follow-up several randomized trials showed reductions in major bleeding, hemorrhagic stroke, and mortality as compared to warfarin [7]. Despite positive clinical outcomes in trials of subjects tolerant of long-term anticoagulation, LAAO is currently only recommended for patients with AF who are unable to tolerate long-term anticoagulation use due to contraindications, high bleeding risk in general, low drug tolerance or low drug adherence [8] at a Class IIb Level B recommendation [6].

A large proportion of real-world LAAO patients present with significant comorbidities [9–11], which pose a special task to the peri-procedural LAAO team. Additionally, despite increasing operator experience [12,13], the procedure itself still poses a complication risk for a number of patients. All these issues may influence the length of stay (LoS), which is a crucial factor for patient comfort and cost in current restricted health care budgets. In this analysis, we aimed to determine the LoS in patients undergoing LAAO in a large world-wide cohort, identify potential risk factors for a late discharge, and investigate additive adverse events during a 60-day follow-up period.

## Methods

Patients with non-valvular AF undergoing LAAO implant attempt within the Amplatzer™ Amulet™ Occluder Observational Study from 06/2015 to 09/2016 were included. The full study design can be found in the S1 File. Length of stay was not a pre-specified endpoint in the study, but we are investigating length of stay as part of a post-hoc sub-analysis. This project was presented and accepted by the scientific board. Participants were classified according to the LoS, with same day (S, 0day), early (E, 1day), regular (R, 2-3days) and late discharge (L, ≥4days after the index procedure) groups defined. Baseline, procedure-related characteristics and serious adverse events (SAE) during the in-hospital stay were analyzed and follow-up through 60 days reviewed. Fig 1 displays the patient stratification process for this sub-analysis.

SAE reporting followed the ISO 141555 definition. Reportable events included procedure and device associated events, major bleeding (Bleeding Academic Research Consortium

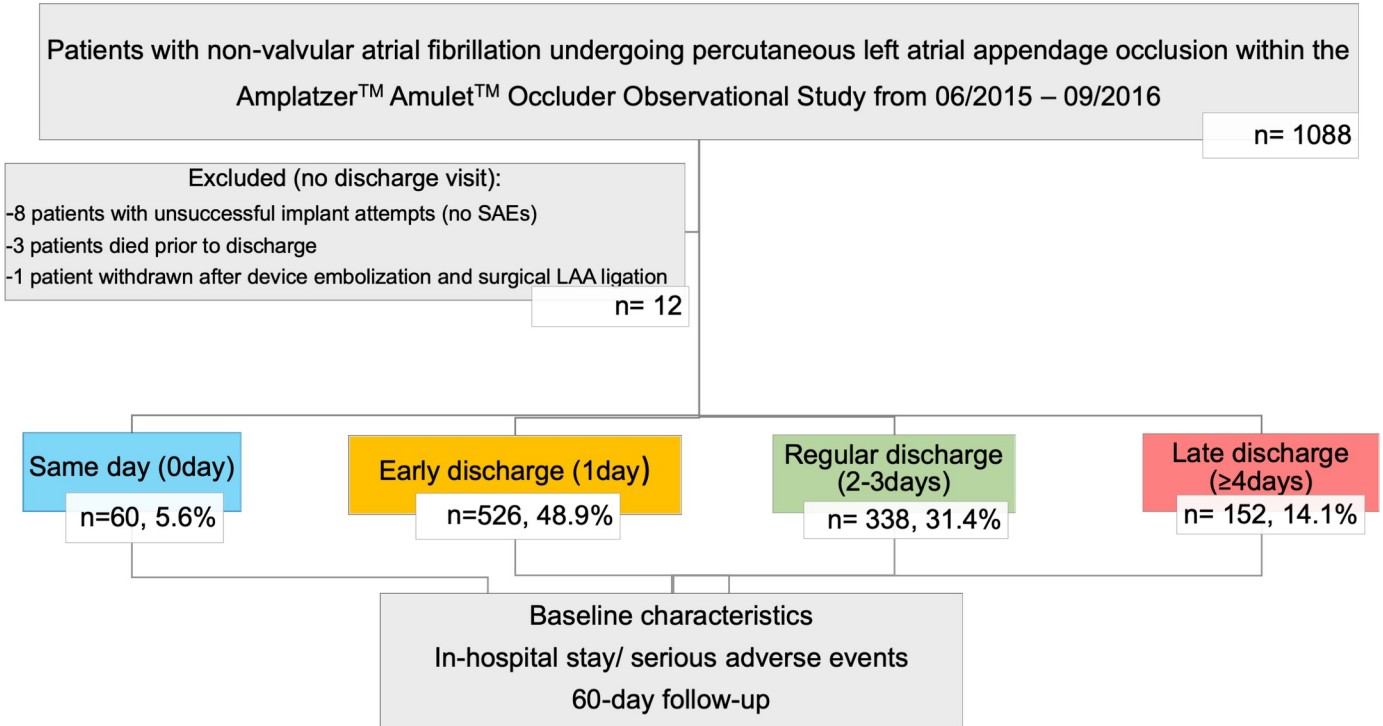

**Fig 1. Patient selection process.** Of 1088 patients undergoing left atrial appendage occlusion within the Amplatzer™ Amulet™ Occluder Observational Study from 06/2015–09/2016, twelve patients had no discharge visit and were excluded for reasons detailed in the Results section. The remaining study population was divided into four length of stay groups and further analyzed.

(BARC) type 3 or greater), neurological deficits such as stroke or transient ischemic attack, and respiratory complications. All SAEs were reviewed by an independent clinical events committee which adjudicated relatedness to the implant procedure and device. SAE were grouped into cardiac (pericardial effusion, pericardial tamponade, device embolization, device thrombus, heart failure), bleeding (access-site hematoma, gastrointestinal bleeding, anemia and subdural hematoma), neurologic (ischemic stroke, seizure), respiratory (pneumoniae, exacerbated chronic obstructive lung disease, respiratory failure, pulmonary embolism) and other events (delirium, urinary retention, transesophageal echocardiography related event, air embolism, arteriovenous fistula and pseudoaneurysm).

## Statistical analysis

Descriptive statistics summarized baseline and procedural characteristics. The Kruskal-Wallis test was used to identify differences in continuous variables between the same day, early, regular and late discharge groups. Fischer's Exact Test was used to identify differences in categorical variables. Multinomial logistic analyses were used to identify risk factors for a prolonged in-hospital stay. The rates of all-cause mortality, ischemic stroke, and major bleeding were calculated at 60-days post-LAAO using Kaplan-Meier method, with the Log-Rank test identifying differences between groups. A hierarchical model adding center effects was not appropriate due to a large number of centers enrolling patients and the sparseness of the data.

A p-value <0.05 was considered statistically significant. SAS 9.4 (SAS Institute, Cary, North Carolina) was used for analysis, STATA/SE 16.0 (StataCorp, College Station, Texas) and Prism 8 (Graphpad Software Inc., San Diego, California) for graphing.

### Ethical standards

This study has been approved by the appropriate ethics committees and has therefore been performed in accordance with the ethical standards laid down in the 1964 Declaration of Helsinki and its later amendments. All persons gave their written informed consent prior to their inclusion in the study. Details that might disclose the identity of the subjects under study are omitted. A full list of institutional review boards, which approved the study, can be found in the S2 File.

## Results

From the initial study population of 1088 patients, 12 were excluded because no discharge visit was performed: 8 with unsuccessful implant attempts (no SAEs), three who died prior to discharge, and one withdrawn after device embolization and surgical LAA ligation. The remaining participants were divided into a same day (0 days, n = 60) an early (1day, n = 526), regular (2-3days, n = 338) and a late discharge (≥4 days after the index procedure, n = 152 **Fig 2**) group and further analyzed.

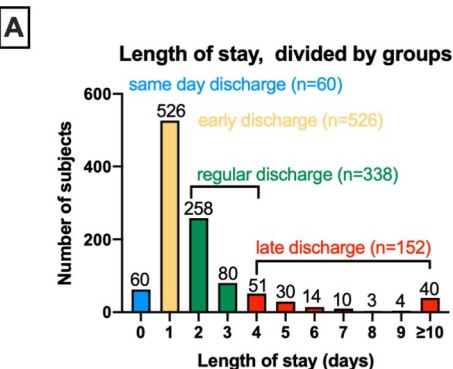

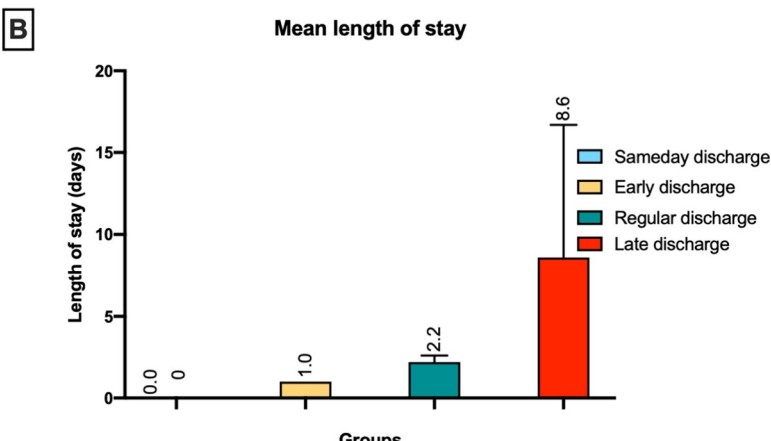

**Fig 2. Length of stay (LoS) after LAAO. (A)** Distribution of patients undergoing percutaneous LAAO within the Amplatzer[TM] Amulet[TM] Occluder Observational Study. **(B)** The mean LoS was 2.2days and 8.6days in the regular and late discharge groups, respectively.

**Table 1. Baseline characteristics.**

| Characteristic | Same Day Discharge (n = 60) | Early Discharge (n = 526) | Regular Discharge (n = 338) | Late Discharge (n = 152) | p-value |
|---|---|---|---|---|---|
| Age (years) | 77±7 | 75±8 | 74±9 | 75±8 | 0.28 |
| Male gender | 73% | 66% | 64% | 57% | 0.08 |
| Atrial fibrillation at time of implant | 75% | 61% | 53% | 60% | <0.01 |
| Hypertension | 82% | 83% | 86% | 84% | 0.45 |
| Congestive heart failure | 10% | 18% | 14% | 25% | 0.01 |
| Previous stroke | 40% | 29% | 25% | 25% | 0.12 |
| Previous TIA | 18% | 10% | 10% | 10% | 0.27 |
| Previous major bleed | 77% | 71% | 72% | 70% | 0.80 |
| Abnormal renal function | 8% | 14% | 14% | 21% | 0.07 |
| Abnormal liver function | 3% | 5% | 6% | 7% | 0.53 |
| Chronic obstructive pulmonary disease | 12% | 10% | 11% | 15% | 0.37 |
| Previous PCI or CABG | 32% | 24% | 23% | 36% | <0.01 |
| Peripheral vascular disease | 7% | 13% | 18% | 19% | 0.02 |
| CHA$_2$DS$_2$-VASc Score | 4.3±1.6 | 4.1±1.5 | 4.1±1.6 | 4.5±1.6 | 0.09 |
| HAS-BLED Score | 3.3±0.9 | 3.3±1.1 | 3.3±1.1 | 3.5±1.2 | 0.10 |
| Contraindication to oral anticoagulation | 73% | 81% | 85% | 87% | 0.04 |
| • Absolute contraindication | 3% | 7% | 7% | 7% | |
| • Relative contraindication | 30% | 34% | 35% | 32% | |
| • Known bleeding risk | 40% | 40% | 43% | 49% | |

CABG: Coronary artery bypass grafting.

PCI: Percutaneous coronary intervention.

TIA: Transitory ischemic attack.

The top three enrolling countries were Germany (n = 378, mean LoS 2.7 days), Italy (n = 178, mean LoS 3.0 days) and Spain (n = 105, mean LoS 3.0 days).

Patients were pre-dominantly male (S: 73% vs. E: 67% vs. R: 64% vs. L: 57%, p = 0.08) and mean age in the mid-seventies (S: 77 vs. E: 75 vs. R: 74: vs. L: 75 years, p = 0.28). The CHA$_2$DS$_2$-VASc (S: 4.3±1.6 vs. E: 4.1±1.5 vs. R: 4.1±1.6 vs. L: 4.5±1.6, p = 0.09) and HAS-BLED scores (S: 3.3±0.9 vs. E: 3.3±1.1 vs. R: 3.3±1.1 vs. L: 3.5±1.2, p = 0.10) were highest in the late discharge group. In addition to increased CHADS-VASc and HAS-BLED scores, the late discharge group more often had history of congestive heart failure, peripheral vascular disease, and abnormal renal function. Further information can be obtained from **Table 1**.

In the late discharge group, procedure- or device-related SAEs occurred in the highest numbers (S: 0.0% vs. E: 1.0% vs. R: 2.1% vs. L: 23.0%, p<0.0001) within seven days. Further procedural and in-hospital outcomes are shown in **Table 2**.

A sub-analysis of the late discharge group revealed that 37 subjects had a SAE (n = 46 events), of those, seven patients with more than one SAE during the hospital course. Of the 37 subjects with SAE prior to discharge, cardiac or bleeding complications were the most common underlying reason for a prolonged LoS (Fig 3), occurring in 26 subjects. Most SAEs (n = 41/46 events, 89.1%) of the late discharge group were adjudicated as related to the implant procedure and/or device. Fifteen of the 16 cardiac SAEs were procedure- or device-related, including 10 cases of pericardial effusion/tamponade. One cases of each AF, acute pulmonary edema, pleural effusion, device embolization, and thrombus on device were the remaining six procedure- or device-related cardiac SAEs.

**Table 2. Procedure related characteristics and serious adverse events during the in-hospital stay.**

| Characteristic | Same Day Discharge (n = 60) | Early Discharge (n = 526) | Regular Discharge (n = 338) | Late Discharge (n = 152) | p-value |
|---|---|---|---|---|---|
| **Echocardiographic guidance** | | | | | |
| Tranoesophageal echocardiography | 98% | 85% | 89% | 91% | <0.01 |
| Intracardiac echocardiography | 2% | 15% | 11% | 9% | |
| Procedure duration (minutes) | 36±25 | 34±22 | 32±22 | 33±23 | 0.23 |
| Maximum ACT (seconds) | 290±175 | 283±99 | 292±92 | 317±111 | <0.01 |
| Total contrast (mL) | 116±80 | 98±76 | 97±80 | 118±96 | 0.03 |
| Total fluoroscopic time (minutes) | 12±6 | 13±11 | 12±11 | 15±11 | <0.01 |
| Technical successful | 100.0% | 99.8% | 99.7% | 100.0% | 1.00 |
| **Serious adverse events (SAE)** | | | | | |
| Procedure-/Device-related SAE ≤1 day | 1.7% | 1.1% | 2.1% | 19.7% | <0.001 |
| Procedure-/Device-related SAE ≤7 days | 1.7% | 2.1% | 4.1% | 22.4% | <0.001 |
| Procedure-/Device-related SAE prior to discharge | 0.0% | 1.0% | 2.1% | 23.0% | <0.001 |
| Ischemic stroke | 0.0% | 0.0% | 0.0% | 0.0% | |
| Myocardial infarction | 0.0% | 0.0% | 0.0% | 0.0% | |
| Pericardial effusion without tamponade | 0.0% | 0.0% | 0.9% | 2.0% | |
| Pericardial effusion with tamponade | 0.0% | 0.4% | 0.0% | 4.6% | |
| Device embolization | 0.0% | 0.0% | 0.0% | 0.7% | |
| Vascular access site complication | 0.0% | 0.4% | 0.6% | 5.3% | |
| TEE-related complication | 0.0% | 0.0% | 0.0% | 1.3% | |
| Pneumothorax | 0.0% | 0.0% | 0.0% | 0.0% | |
| **Length of stay (days)** | 0.0±0.0 | 1.0±0.0 | 2.2±0.4 | 8.6±8.2 | <0.001 |

ACT: Activated clotting time.

Univariable multinomial logistic analysis (length of stay categorized into 4 categories: 0 day, 1 day, 2–3 days, and ≥4 days) only identified HAS-BLED score as an independent influencing factor (p = 0.04). Results for gender (Female vs Male) and $CHA_2DS_2$-VASc score were not statistically significant (p = 0.08 and 0.09 respectively). As shown in **Table 3**, the odds ratio for late discharge relative to early discharge subjects is 1.271 for every one unit increase in HAS-BLED score, indicating an association between higher HAS-BLED score and longer

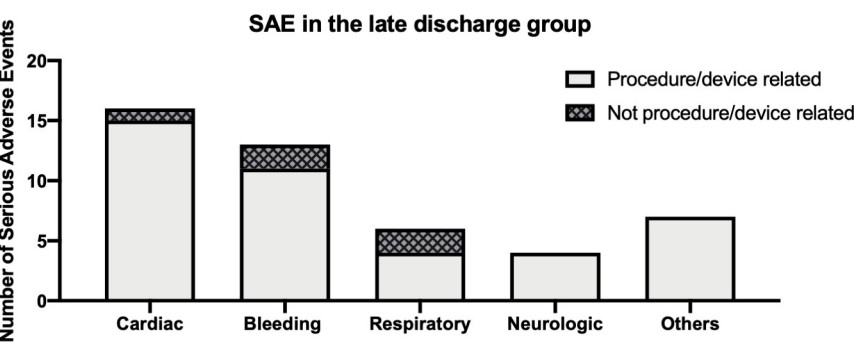

**Fig 3. Serious adverse events (SAE) during the in-hospital stay of the late discharge group.** Overall, 46 SAEs occurred in 37 late discharge group subjects. Most common underlying reasons were cardiac complications (34.8%), followed by bleeding complications (32.6%). Respiratory (19.6%) and neurological events (9.3%) were of minor importance.

**Table 3. Multinomial logistic analysis of length of stay with HAS-BLED score.**

| | | Odds Ratio Estimates | | |
|---|---|---|---|---|
| Effect | Length of Stay | Point Estimate | 95% Wald Confidence Limits | |
| HAS-BLED Score | Late Discharge | 1.270 | 1.078 | 1.497 |
| HAS-BLED Score | Regular Discharge | 1.032 | 0.910 | 1.172 |
| HAS-BLED Score | Same Day Discharge | 1.057 | 0.827 | 1.352 |

Early discharge is the reference timeframe other length of stay timeframes are compared to.

length of stay following LAAO. As illustrated in Table 1, the HAS-BLED score was similar for patients with same day, early, or regular discharge durations (mean of 3.3), but elevated in the late discharge group (mean of 3.5).

During a 60-day follow-up period, the overall mortality (S: 0.0% vs. E: 1.0% vs. R: 1.2% vs. L: 3.3%, p = 0.106, **Fig 4A**) was comparable. The rate of major bleeding events (S: 0.0% vs. E: 5.2% vs. R: 2.7% vs. L: 16.5%, p<0.0001, **Fig 4B**) and ischemic stroke (S: 0.0% vs. E: 0.4% vs. R: 0.3% vs. L: 4.0%, p<0.0001, **Fig 4C**) was the highest in the late discharge group.

Events after discharge and through 60 days post-procedure were reviewed. Even after discharge, the risk for the occurrence of a SAE was increased in the late discharge group. Death after discharge occurred in 5 patients (S:0 (0.0%) vs. E: 5 (1.0%) vs. R: 4 (1.2%) vs. L: 5 (3.3%), p = 0.160), 8 major bleeding events were reported in 7 patients (S: 0 (0.0%) vs. E: 23 (4.4%) vs. R: 8 (2.4%) vs. L: 7 (4.6%), p = 0.160) and ischemic stroke occurred in 3 patients (S: 0 (0.0%) E: 2 (0.4%) vs. R: 1 (0.3%) vs. L: 3 (2.0%), p = 0.134).

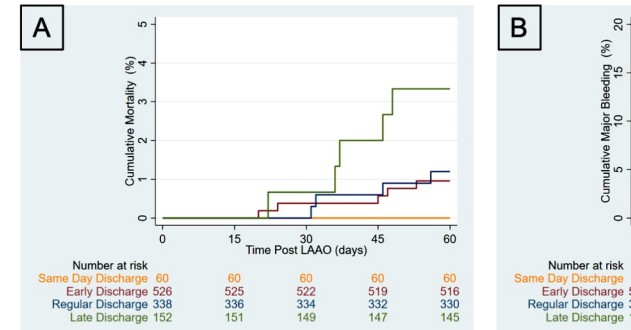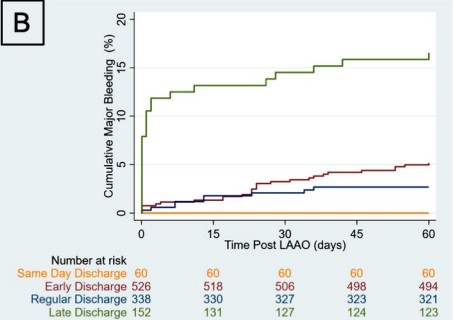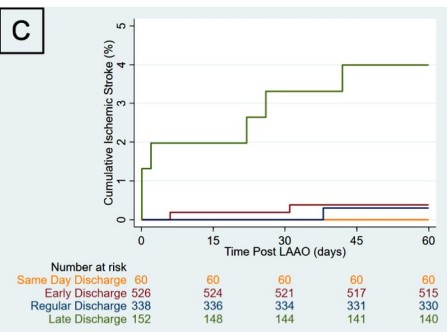

| | Mortality | Major bleeding | Ischemic stroke |
|---|---|---|---|
| **Sameday discharge** | 0.0% | 0.0% | 0.0% |
| **Early discharge** | 1.0% | 5.2% | 0.4% |
| **Regular discharge** | 1.2% | 2.7% | 0.3% |
| **Late discharge** | 3.3% | 16.5% | 4.0% |
| **p-value** | 0.106 | <0.0001 | <0.0001 |

**Fig 4. 60-day follow-up. (A)** The overall mortality (S: 0.0% vs. E: 1.0% vs. R: 1.2% vs. L: 3.3%, p = 0.106) was not different at 60-days. (**B** and **C**) The rate of major bleeding (S: 0.0% vs. E: 5.2% vs. R: 2.7% vs. L: 16.5%, p<0.0001) events and ischemic stroke (S: 0.0% vs. E: 0.4% vs. R: 0.3% vs. L: 4.0%, p<0.0001) was the highest in the late discharge group.

## Discussion

In the field of structural heart disease, percutaneous LAAO has emerged as a guideline recommended stroke prevention strategy performed with an increasing case load worldwide [14–16].

To our knowledge, a detailed LoS analysis, in a large number of patients from multiple geographies, has not yet been performed. Our study shows that **1)** SAE are a major trigger for a prolonged LoS **2)** the most common underlying reasons are cardiac and bleeding complications and **3)** only the HAS-BLED score is independently associated with a longer in-hospital stay.

### SAE as a major trigger for a prolonged LoS

LoS analyses of LAAO patients were previously performed in small numbers and mostly concentrate on the safety and feasibility of early or same-day discharge concepts [17–19] since almost all health-care systems have to cope with budget-restrictions. These concepts are applicable for low risk patients with a high likelihood for an uneventful procedure. Although infrequent, SAEs associated with LAAO procedures or devices may be associated with increased mortality and morbidity [20]. Patients undergoing LAAO usually present with significant comorbidities, which may contribute to the occurrence of intraprocedural events. The incidence of peri-procedural SAEs in this global registry is in line with results from other recent LAAO registries, which is a positive result for a rather "early" experience with percutaneous LAAO [21,22]. For the first time, we report a strong correlation of SAEs with prolonged LoS. The peri-procedural phase specifically demands attention. Prompt identification and close monitoring of SAE during that period is crucial for patient safety and comfort and may reduce the LoS and associated resources and expenses. While bleeding and cardiac complications are potentially dangerous, this study suggests that proper care can be provided during the in-hospital setting. These complications may be in part due to the high intrinsic bleeding risk of the LAAO population studied and an extended LoS may be required to deal with the complications. However, even after discharge, patients with a prolonged intra-hospital LoS tended to experienced more SAEs, indicating that a late discharge may define an at-risk patient group, requiring closer monitoring after the initial, interventional phase.

Despite SAEs as a major trigger for a prolonged LoS, we see a difference in LoS between the United Kingdom and Nordic states compared to the rest of Europe (i.e., Germany, Italy, France and Spain). This might be attributed to country-based differences in reimbursement, hospital market competition and procedural volume, but cannot be proven with the current data set.

### HAS-BLED score as a predictive factor for prolonged LoS after LAAO

The HAS-BLED score was developed as a practical risk score to predict major bleeding events in patients on oral anticoagulation and hence, enables a benefit-risk estimation in patients with atrial fibrillation [23,24]. It has also been used in percutaneous coronary intervention studies to predict major bleeding events [25] and survival in patients without AF [26]. The score combines modifiable and unmodifiable pre-conditions of the patient and clinicians should revisit the HAS-BLED score to re-confirm correctable risk factors to avoid bleeding and, as our study could show, a prolonged length of stay in patients undergoing percutaneous LAAO. For example, if uncontrolled hypertension is present, the blood pressure level should be improved prior to device implantation. In case of warfarin intake and labile INR, patients may be switched to unfractionated heparine prior to the procedure to avoid inadequate blood-thinning. Those lifestyle modifications and/or drug adjustments may delay the procedure but

have the potential to significantly improve outcome. In case of non-modifiable risk factors leading to a high HAS-BLES score, the implanting team has to be aware of an increased risk for major complications and subsequently a prolonged LoS after device implantation.

### Same day discharge versus early discharge

Looking at the same day discharge and early discharge groups we see no differences. Therefore, it seems reasonable to base the decision upon whether same day or early discharge is performed on patient wish and social environment. A successful procedure without any SAE often occurred for patients discharged within either time frame and may be the primary factor for allowing same day or early discharge.

### Limitations

This is a prospective, multicenter-global trial with over 1000 participants. However, it is not randomized and only one specific device for LAAO was utilized. The study was initially not powered to determine risk factors for the LoS after LAAO. Discharge policy may vary by institution and country in aspects that are unknown or that we are unable to adjust for. A detailed cost analysis in the participating countries is not able to be performed with the study dataset. The study refers to an "early" experience with percutaneous LAAO (2015–2016) with a single device and may not be representative for current practice.

### Conclusion

Over half of the subjects receiving an Amplatzer Amulet occluder were discharged within 1 day of the implant procedure. HAS-BLED score was the only independent risk factor associated with a prolonged in-hospital stay. SAE are a major trigger for a late discharge after LAAO, with rates of major bleeding and ischemic stroke significantly greater in the late discharge group. The 60-day mortality rate did not differ between patients grouped by length of stay.

### Supporting information

**S1 Checklist. TREND statement checklist.**
(PDF)

**S1 File. AMPLATZER™ Amulet™ Observational post-market study clinical protocol.**
(PDF)

**S2 File. List of all institutional review boards which approved the study.**
(PDF)

### Author Contributions

**Conceptualization:** Kerstin Piayda, Shazia Afzal, Jens Erik Nielsen-Kudsk, Patrizio Mazzone, Matteo Montorfano, Tobias Zeus.

**Data curation:** Matteo Montorfano, Tobias Zeus.

**Formal analysis:** Kerstin Piayda, Tobias Zeus.

**Investigation:** Kerstin Piayda, Boris Schmidt, David Hildick-Smith, Tobias Zeus.

**Methodology:** Kerstin Piayda, Tobias Zeus.

**Project administration:** Tobias Zeus.

**Resources:** Ryan Gage, Tobias Zeus.

**Supervision:** Jens Erik Nielsen-Kudsk, Tobias Zeus.

**Validation:** Hong Zhao.

**Visualization:** Ryan Gage, Tobias Zeus.

**Writing – original draft:** Kerstin Piayda, Tobias Zeus.

**Writing – review & editing:** Shazia Afzal, Jens Erik Nielsen-Kudsk, Boris Schmidt, Patrizio Mazzone, Sergio Berti, Sven Fischer, Juha Lund, Matteo Montorfano, David Hildick-Smith, Ryan Gage, Tobias Zeus.

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
