## [Decision Letter · Decision Letter 0]

21 Dec 2020

PONE-D-20-26960

Length of stay following percutaneous left atrial appendage occlusion: data from the prospective, multicenter Amplatzer™ Amulet™ Occluder Observational Study

PLOS ONE

Dear Dr. Zeus,

Thank you for submitting your manuscript to PLOS ONE. After careful consideration, we feel that it has merit but does not fully meet PLOS ONE’s publication criteria as it currently stands. Therefore, we invite you to submit a revised version of the manuscript that addresses the points raised during the review process.

Please take into great consideration Reviewers' comments about availability of data, according to Plos ONE editorial policy, about the need of re-grouping patients and changing statistical analysis accordingly, about randomized data in favor of LAAO.

We look forward to receiving your revised manuscript.

Kind regards,

Giuseppe Andò, M.D., Ph.D.

Academic Editor

PLOS ONE

Journal Requirements:

2. Please provide additional details on the Clinical Trial study design within the Methods section and ensure that you have provided sufficient information to allow your work to be replicated.

Furthermore, thank you for providing the statement “Length of stay was not a pre-specified endpoint in the study, but we are investigating length of stay as part of a post-hoc sub-analysis. This project was presented and accepted by the scientific board.” in response to our query regarding the discrepancy in the study outcome. Please also provide this statement of clarification in the main manuscript text.

3.Thank you for including your ethics statement:  "    Ethical standards: This study has been approved by the appropriate ethics committee and has therefore been performed in accordance with the ethical standards laid down in the 1964 Declaration of Helsinki and its later amendments. All persons gave their informed consent prior to their inclusion in the study. Details that might disclose the identity of the subjects under study are omitted.".   

5.Thank you for stating the following in the Financial Disclosure section:

We note that one or more of the authors are employed by a commercial company: Harzklinikum Dorothea Christiane Erxleben GmbH

6. We note that you have indicated that data from this study are available upon request. PLOS only allows data to be available upon request if there are legal or ethical restrictions on sharing data publicly. For more information on unacceptable data access restrictions, please see http://journals.plos.org/plosone/s/data-availability#loc-unacceptable-data-access-restrictions.

7.We note that [Figure(s) 3] in your submission contain map images which may be copyrighted. All PLOS content is published under the Creative Commons Attribution License (CC BY 4.0), which means that the manuscript, images, and Supporting Information files will be freely available online, and any third party is permitted to access, download, copy, distribute, and use these materials in any way, even commercially, with proper attribution. For these reasons, we cannot publish previously copyrighted maps or satellite images created using proprietary data, such as Google software (Google Maps, Street View, and Earth). For more information, see our copyright guidelines: http://journals.plos.org/plosone/s/licenses-and-copyright.

1.    You may seek permission from the original copyright holder of Figure(s) [3] to publish the content specifically under the CC BY 4.0 license. 

Reviewers' comments:

Reviewer's Responses to Questions

**Comments to the Author**

1. Is the manuscript technically sound, and do the data support the conclusions?

Reviewer #1: Yes

Reviewer #2: Partly

Reviewer #3: Yes

2. Has the statistical analysis been performed appropriately and rigorously? 

Reviewer #1: Yes

Reviewer #2: No

Reviewer #3: Yes

3. Have the authors made all data underlying the findings in their manuscript fully available?

Reviewer #1: Yes

Reviewer #2: No

Reviewer #3: Yes

4. Is the manuscript presented in an intelligible fashion and written in standard English?

Reviewer #1: Yes

Reviewer #2: Yes

Reviewer #3: Yes

5. Review Comments to the Author

Reviewer #1: Please revise the statement in the Introduction to the effect that there are randomized data showing that left atrial appendage occlusion (LAAO) is preferable to oral anticoagulation (OAC) in patients without contraindication to oral anticoagulation. There was a mortality benefit in PROTECT-AF at long-term. You may then add that illogically general customs and guidelines recommend the use of LAAO in patients with AF only in those who are unable to tolerate long-term anticoagulation use due to contraindications, high bleeding risk in general, low drug tolerance, or low drug adherence.

The main difference lies between same day procedures and procedures with overnight surveillance. Please make a separate group of "Same day discharge" patients. It is, of course, not important for this study if patients spent the night before the procedure at the hospital or not.

So, your new groups should be: Same day discharge (or Outpatient procedure, meaning 0 night after LAAO; according to Figure 2, there were 62 such patients), early discharge (1 night), Regular discharge (2/3 nights), and Late discharge (≥4 nights).

This will unfortunately entail to change most statistics, tables, and figures.

What do you mean by "12 patients had no discharge visit and were excluded"? Did they die in the hospital or do you simply not know how long they stayed?

In table 2, there are no figures for Intracardiac echocardiography. Looking at Total echocardiography and Transesophageal echocardiography, they seem to account for the difference, i.e., the majority of patients.

Please provide a Table 4 with specific complications (e.g., cerebral ischemia, myocardial infarction, pericardial bleed, cardiac tamponade, device embolization, puncture site problem, bronchial aspiration or other problems of transesophageal echocardiography, pneumothorax due to central line, etc.).

At the end, please come up with some recommendations what supports same day discharge and what mandates at least 1 night at the hospital after LAAO (e.g., chest pain, pericardial effusion at echocardiography, etc.). So, readers can benefit from the paper.

Reviewer #2: The manuscript addresses an interesting topic. The data have some specific features deserving to be analyzed. The research questions may lead to improvements in the existing literature. Nevertheless, there are several methodological/statistical issues which must be addressed to ensure the reliability of the results.

1. The data are not fully available. This is not in line with the journal's guidelines (https://journals.plos.org/plosone/s/data-availability) and does not allow the reviewer to reproduce the results. Please, upload the data and the code used to obtain the results.

2. The statistical methods employed should be better described. At first glance, a linear regression model is considered to model the length of stay (LOS). However, the LOS is a generally a count variable; isn't it? As counts, the correct starting point model is the Poisson regression. Please, be aware that any models is based on some assumptions which must be fulfilled to ensure the reliability of the results. For example, the linear regression must satisfy the Gauss-Markov assumptions, the Poisson regression is based on the equidispersion assumption, etc. Accordingly, please, amend the regression part choosing for the regression model according to the nature and characteristics of the dependent variable at hand.

3. The study is multicenter and, as such, observations belonging to the same center may be correlated. In other words, the data structure is hierarchical. This hierarchy should be accounted for in the modelling, i.e. heterogeneity across centers must be addressed.

4. A more detailed survival analysis would be appreciated. A proportional hazard Cox model may be considered (again, please, check for model's assumptions) and some comments on the main variables affecting mortality may be discussed.

5. Please, provide evidence that the assumptions behind the parametric tests implemented to obtain results in Table 1 are fulfilled. If they are not, please, move to non-parametric testing procedures.

Reviewer #3: The article "Length of stay following percutaneous LAAO: data from the prospective, multicenter Amplatzer Amulet Occluder Observational Study", reports data regarding the influence of LAAO (with Amplatzer Amulet) on lenght of in-hospital stay, in a population of patients treated between 2015 and 2016.

The manuscript is well written in good English, data are elegantly analyzed and reported by the Authors.

Limitations are well described in the dedicate session.

Minor point:

- Manuscript preparation should be revised, since Figure legends are included in the manuscript (rather than listed at the end), making article revision more difficult.

Major point:

- The Study refers to a "early" experience with percutaneous LAAO (2015-2016) with a single "first-generation" device; this should be discussed in either positive or negative way (considering the results), due to the influence of procedural experience on both ischemic and hemorragic adverse events.

6. PLOS authors have the option to publish the peer review history of their article (what does this mean?). If published, this will include your full peer review and any attached files.

Reviewer #1: No

Reviewer #2: No

Reviewer #3: No

---

## [Author Response · Author response to Decision Letter 0]

9 Mar 2021

Journal Requirements:

https://journals.plos.org/plosone/s/file?id=wjVg/PLOSOne_formatting_sample_main_body.pdfand

• Reply: We went through the PLOS ONE’s style requirements and updated the manuscript and file names accordingly.

2. Please provide additional details on the Clinical Trial study design within the Methods section and ensure that you have provided sufficient information to allow your work to be replicated.

Furthermore, thank you for providing the statement “Length of stay was not a pre-specified endpoint in the study, but we are investigating length of stay as part of a post-hoc sub-analysis. This project was presented and accepted by the scientific board.” in response to our query regarding the discrepancy in the study outcome. Please also provide this statement of clarification in the main manuscript text.

• Reply: We added this statement to our method section and provide the full study design as supplementary file S1. Please see p. 4 l.78-82 and supplementary file S1

3.Thank you for including your ethics statement: " Ethical standards: This study has been approved by the appropriate ethics committee and has therefore been performed in accordance with the ethical standards laid down in the 1964 Declaration of Helsinki and its later amendments. All persons gave their informed consent prior to their inclusion in the study. Details that might disclose the identity of the subjects under study are omitted.". 

• Reply: We included a supplement file with all review boards which approved the study. (Supplementary file S2)

5.Thank you for stating the following in the Financial Disclosure section:

We note that one or more of the authors are employed by a commercial company: Harzklinikum Dorothea Christiane Erxleben GmbH

• Reply: Harzklinikum Dorothea Christiane Erxleben GmbH is not a funder of the study but a commercial company. It provided support in the form of salaries for the author S.F., but did not have any additional role in the study design, data collection and analysis, decision to publish, or preparation of the manuscript

• Reply: The funder (Abbott) provided support in the form of salaries for authors R.G. and H.Z., but did not have any additional role in the study design, data collection and analysis, decision to publish, or preparation of the manuscript.

• Reply: Dr. Nielsen-Kudsk has served as a proctor for Abbott and is a consultant for Abbott and Boston Scientific. Dr. Schmidt has served as a consultant for Boston Scientific and Medtronic. Dr. Mazzone has served as a consultant for Abbott, Boston Scientific, and Medtronic. Dr. Berti has served as a proctor for Abbott. Dr. Fischer has served as a proctor for Biotronik and Boston Scientific and is a consultant for Abbott. Dr. Lund has served as a proctor for Abbott. Dr. Montorfano has served as a proctor for Abbott, Boston Scientific, and Edwards Lifesciences. Dr. Hildick-Smith has served as a proctor and consultant for Abbott. Mr. Gage and Ms. Hong Zhao are employees of Abbott (funder of the initial study). Dr. Zeus has received consulting fees, travel expenses or study honorariums from Medtronic and Edwards Lifesciences outside of this work. All other authors have nothing to disclose with regard to this project.

 Within your Competing Interests Statement, please confirm that this commercial affiliation does not alter your adherence to all PLOS ONE policies on sharing data and materials by including the following statement: "This does not alter our adherence to PLOS ONE policies on sharing data and materials.” (as detailed online in our guide for authorshttp://journals.plos.org/plosone/s/competing-interests) . If this adherence statement is not accurate and there are restrictions on sharing of data and/or materials, please state these. Please note that we cannot proceed with consideration of your article until this information has been declared.

• Reply: No data are available on a public platform. Individual deidentified participant data cannot be shared. The sponsor of this study (Abbott) is unable to legally distribute/provide access to the study data at the time of publication due to privacy concern. However, Abbott provides contact information instead where an interested researcher could apply to gain access to the data

Website: https://www.cardiovascular.abbott/us/en/hcp/investigator-sponsored-studies.html

Email: abbottissrequests@abbott.com

• Reply: We did include an updated Funding and Competing Interest Statement in our cover letter.

6. We note that you have indicated that data from this study are available upon request. PLOS only allows data to be available upon request if there are legal or ethical restrictions on sharing data publicly. For more information on unacceptable data access restrictions, please see http://journals.plos.org/plosone/s/data-availability#loc-unacceptable-data-access-restrictions.

• Reply: No data are available on a public platform. Individual deidentified participant data cannot be shared. The sponsor of this study (Abbott) is unable to legally distribute/provide access to the study data at the time of publication due to privacy concern. However, Abbott provides contact information instead where an interested researcher could apply to gain access to the data

Website: https://www.cardiovascular.abbott/us/en/hcp/investigator-sponsored-studies.html

Email: abbottissrequests@abbott.com

• Reply: Thank you. An updatedversion of the data availability statement can be found in the cover letter.

7.We note that [Figure(s) 3] in your submission contain map images which may be copyrighted. All PLOS content is published under the Creative Commons Attribution License (CC BY 4.0), which means that the manuscript, images, and Supporting Information files will be freely available online, and any third party is permitted to access, download, copy, distribute, and use these materials in any way, even commercially, with proper attribution. For these reasons, we cannot publish previously copyrighted maps or satellite images created using proprietary data, such as Google software (Google Maps, Street View, and Earth). For more information, see our copyright guidelines: http://journals.plos.org/plosone/s/licenses-and-copyright.

• Reply: We decided to remove Fig 3

Reviewers' comments:

Reviewer's Responses to Questions

Comments to the Author

1. Is the manuscript technically sound, and do the data support the conclusions?

Reviewer #1: Yes

Reviewer #2: Partly

Reviewer #3: Yes

2. Has the statistical analysis been performed appropriately and rigorously?

Reviewer #1: Yes

Reviewer #2: No

Reviewer #3: Yes

3. Have the authors made all data underlying the findings in their manuscript fully available?

Reviewer #1: Yes

Reviewer #2: No

Reviewer #3: Yes

4. Is the manuscript presented in an intelligible fashion and written in standard English?

Reviewer #1: Yes

Reviewer #2: Yes

Reviewer #3: Yes

5. Review Comments to the Author

Reviewer #1: Please revise the statement in the Introduction to the effect that there are randomized data showing that left atrial appendage occlusion (LAAO) is preferable to oral anticoagulation (OAC) in patients without contraindication to oral anticoagulation. There was a mortality benefit in PROTECT-AF at long-term. You may then add that illogically general customs and guidelines recommend the use of LAAO in patients with AF only in those who are unable to tolerate long-term anticoagulation use due to contraindications, high bleeding risk in general, low drug tolerance, or low drug adherence.

• Reply: In the introduction we stated that “However, percutaneous left atrial appendage occlusion (LAAO) has emerged as a favorable non-pharmacological stroke prevention strategy in patients with AF who are unable to tolerate long-term anticoagulation use due to contraindications, high bleeding risk in general, low drug tolerance or low drug adherence”. We did never emphasize that LAAO is preferable to OAC in patients without contraindications to oral anticoagulation.

The main difference lies between same day procedures and procedures with overnight surveillance. Please make a separate group of "Same day discharge" patients. It is, of course, not important for this study if patients spent the night before the procedure at the hospital or not.

So, your new groups should be: Same day discharge (or Outpatient procedure, meaning 0 night after LAAO; according to Figure 2, there were 62 such patients), early discharge (1 night), Regular discharge (2/3 nights), and Late discharge (≥4 nights).

• Reply: According to the reviewers’ recommendations we did rearrange the grouping in patients with same-day discharge, early discharge, regular discharge and late discharge. Results are presented within the text, figures and tables accordingly throughout the whole manuscript. 

This will unfortunately entail to change most statistics, tables, and figures.

What do you mean by "12 patients had no discharge visit and were excluded"? Did they die in the hospital or do you simply not know how long they stayed?

• Reply: Thank you for the opportunity to clarify. We went back to the original data and identified the reasons for exclusion: 8 patients had unsuccessful implant attempts (no SAEs), three died prior to discharge, and one was withdrawn after device embolization and surgical LAA ligation. We added this information in our result section and to Fig. 1. Please see Fig. 1 and p. 6 l. 128-130

In table 2, there are no figures for Intracardiac echocardiography. Looking at Total echocardiography and Transesophageal echocardiography, they seem to account for the difference, i.e., the majority of patients.

• Reply: The numbers for intracardiac echocardiography are now implemented in the table. Please see Table 2 (p.8)

Please provide a Table 4 with specific complications (e.g., cerebral ischemia, myocardial infarction, pericardial bleed, cardiac tamponade, device embolization, puncture site problem, bronchial aspiration or other problems of transesophageal echocardiography, pneumothorax due to central line, etc.).

• Reply: We added an additional table to the manuscript and report relevant adverse events related to the LAAO within this study. We embedded this information into Table 2 (p.8-9) after the SAE reporting section.

 Same Day Early Regular Late

 n=62 n=526 n=338 n=150

Ischemic stroke 0.0% 0.0% 0.0% 2.0%

Myocardial infarction 0.0% 0.0% 0.0% 0.0%

Pericardial effusion without tamponade 0.0% 0.0% 0.9% 2.0%

Pericardial effusion with tamponade 1.6% 0.4% 0.0% 4.0%

Device embolization 0.0% 0.0% 0.0% 0.7%

Vascular access site complication 0.0% 0.4% 0.6% 5.3%

TEE-related complication 1.6% 0.0% 0.0% 0.7%

Penumothorax 0.0% 0.0% 0.0% 0.0%

At the end, please come up with some recommendations what supports same day discharge and what mandates at least 1 night at the hospital after LAAO (e.g., chest pain, pericardial effusion at echocardiography, etc.). So, readers can benefit from the paper.

• Reply: Looking at the provided data, this is a tricky question. We do not see a relevant difference in complications during the same day and early discharge group. In this case, we cannot provide useful suggestions for the readers on suggestions for sameday vs. early discharge.

Reviewer #2: The manuscript addresses an interesting topic. The data have some specific features deserving to be analyzed. The research questions may lead to improvements in the existing literature. Nevertheless, there are several methodological/statistical issues which must be addressed to ensure the reliability of the results.

1. The data are not fully available. This is not in line with the journal's guidelines (https://journals.plos.org/plosone/s/data-availability) and does not allow the reviewer to reproduce the results. Please, upload the data and the code used to obtain the results.

• Reply: After discussion with the sponsor, no data are available on a public platform. Individual deidentified participant data cannot be shared. The sponsor of this study (Abbott) is unable to legally distribute/provide access to the study data at the time of publication due to privacy concern. However, Abbott provides contact information instead where an interested researcher could apply to gain access to the data

Website: https://www.cardiovascular.abbott/us/en/hcp/investigator-sponsored-studies.html

Email: abbottissrequests@abbott.com

We updated our data sharing agreement accordingly.

2. The statistical methods employed should be better described. At first glance, a linear regression model is considered to model the length of stay (LOS). However, the LOS is a generally a count variable; isn't it? As counts, the correct starting point model is the Poisson regression. Please, be aware that any models is based on some assumptions which must be fulfilled to ensure the reliability of the results. For example, the linear regression must satisfy the Gauss-Markov assumptions, the Poisson regression is based on the equidispersion assumption, etc. Accordingly, please, amend the regression part choosing for the regression model according to the nature and characteristics of the dependent variable at hand.

• Reply: We thank the reviewer for the suggestion. We agree that Poisson regression is the appropriate model for the count data, in general. However, a limitation of the model is the assumption that the variance = the mean, which may not always be true. For the “length of stay” variable, values range from 0 to 57 days, with much larger variance (15.8) compared to the mean (2.4), indicating overdispersion. Therefore, we reanalyzed the data using a multinomial logistic model, where “length of stay” is categorized into 4 categories: 0 day (“same day discharge”), 1 day, 2-3 days, and ≥4 days as suggested by another reviewer (Reviewer #1) for meaningful clinical interpretation.

Univariate multinomial logistic analysis only identified baseline HAS-BLED score as an independent influencing factor (p=0.04, please find details Below). Results for sex (Female vs Male) and CHA2DS2-VASc score are not statistically significant (p= 0.1 and 0.07, respectively). As shown in Table below, the odds ratio for late discharge relative to early discharge is 1.271 based on a one unit increase in HAS-BLED score, indicating an association between higher HAS-BLED score and longer length of stay following the procedure. These results were updated in the Regression Section of the manuscript accordingly. Please see p. 10 l. 181-189 and the added paragraph in the discussion section (p. 13-14 l. 254-268)

Table 3: Results of Multinomial Logistic Analysis of Length of Stay with HAS-BLED Score (please see p.10/11)

Odds Ratio Estimates

Effect Length of Stay Point Estimate 95% Wald

Confidence Limits

HAS-BLED Score Late Discharge 1.271 1.077 1.499

HAS-BLED Score Regular Discharge 1.032 0.910 1.172

HAS-BLED Score Same Day Discharge 1.063 0.834 1.354

3. The study is multicenter and, as such, observations belonging to the same center may be correlated. In other words, the data structure is hierarchical. This hierarchy should be accounted for in the modelling, i.e. heterogeneity across centers must be addressed.

• Reply: Thanks for the suggestion. A total of 61 centers enrolled subjects. Among those, 14 centers enrolled 5 or less subjects with a total of 44 subjects; 13 centers enrolled between 6 and 10 subjects with a total of 101 subjects. 

Therefore, a hierarchical model adding center effects may not be appropriate here due to a large number of centers and the sparseness of the data. We added this information to our method section. Please see p. 5 l. 112-114

4. A more detailed survival analysis would be appreciated. A proportional hazard Cox model may be considered (again, please, check for model's assumptions) and some comments on the main variables affecting mortality may be discussed.

• Reply: Thank you for this comment. We evaluated the Cox regression model, however, the proportional hazard assumption was not met. Therefore, a Cox regression model analysis was deemed not appropriate. Therefore, we did not change the survival analysis. 

5. Please, provide evidence that the assumptions behind the parametric tests implemented to obtain results in Table 1 are fulfilled. If they are not, please, move to non-parametric testing procedures.

• Reply: Within the “Statistical analysis” sub-section of the Methods section we state the statistical method utilized in Table 1: “The Kruskal-Wallis test was used to identify differences in continuous variables between the early, regular and late discharge group. Fischer’s Exact Test was used to identify differences in categorical variables”. Therefore, this methodology utilized non-parametric testing in Table 1. 

Reviewer #3: The article "Length of stay following percutaneous LAAO: data from the prospective, multicenter Amplatzer Amulet Occluder Observational Study", reports data regarding the influence of LAAO (with Amplatzer Amulet) on lenght of in-hospital stay, in a population of patients treated between 2015 and 2016.

The manuscript is well written in good English, data are elegantly analyzed and reported by the Authors.

Limitations are well described in the dedicate session.

Minor point:

- Manuscript preparation should be revised, since Figure legends are included in the manuscript (rather than listed at the end), making article revision more difficult.

Major point:

• Reply: Thank you for pointing this out. We went through the PLOS One style requirements and formatted the manuscript accordingly.

- The Study refers to a "early" experience with percutaneous LAAO (2015-2016) with a single "first-generation" device; this should be discussed in either positive or negative way (considering the results), due to the influence of procedural experience on both ischemic and hemorragic adverse events.

• Reply: We added a paragraph to the discussion to the discussion section to emphasize the effect of early generation devices on LAAO outcomes. Please see p. 12 l. 235-236 and p. 14 l. 276-278.

---

## [Decision Letter · Decision Letter 1]

6 Apr 2021

PONE-D-20-26960R1

Length of stay following percutaneous left atrial appendage occlusion: data from the prospective, multicenter Amplatzer™ Amulet™ Occluder Observational Study

PLOS ONE

Dear Dr. Zeus,

Thank you for submitting your manuscript to PLOS ONE. After careful consideration, we feel that it has merit but does not fully meet PLOS ONE’s publication criteria as it currently stands. Therefore, we invite you to submit a revised version of the manuscript that addresses the points raised during the review process.

Please carefully address Reviewers' 1 comments regarding the first round of review. 

We look forward to receiving your revised manuscript.

Kind regards,

Giuseppe Andò, M.D., Ph.D.

Academic Editor

PLOS ONE

Reviewers' comments:

Reviewer's Responses to Questions

**Comments to the Author**

1. If the authors have adequately addressed your comments raised in a previous round of review and you feel that this manuscript is now acceptable for publication, you may indicate that here to bypass the “Comments to the Author” section, enter your conflict of interest statement in the “Confidential to Editor” section, and submit your "Accept" recommendation.

Reviewer #1: (No Response)

Reviewer #3: All comments have been addressed

2. Is the manuscript technically sound, and do the data support the conclusions?

Reviewer #1: Yes

Reviewer #3: Yes

3. Has the statistical analysis been performed appropriately and rigorously? 

Reviewer #1: Yes

Reviewer #3: Yes

4. Have the authors made all data underlying the findings in their manuscript fully available?

Reviewer #1: No

Reviewer #3: Yes

5. Is the manuscript presented in an intelligible fashion and written in standard English?

Reviewer #1: Yes

Reviewer #3: Yes

6. Review Comments to the Author

Reviewer #1: The amendments are in part adequate. Yet, you need to dwell more on your new group of same day discharge patients which is the most interesting group of all.

Moreover, you completely misread my initial comment. Your answer is the opposite of what I expected and intended to achieve. Please read my statement more carefully and amend your text accordingly: After 2 randomized trials showing a mortality benefit of left atrial appendage closure (LAAC) over oral anticoagulation (OAC) which materializes after 2 years and becomes more and more conspicuous with time, you cannot (cannot!) state that "However, percutaneous left atrial appendage occlusion (LAAO) has emerged as a favorable non pharmacological stroke prevention strategy in patients with AF who are unable to tolerate long-term anticoagulation use due to contraindications, high bleeding risk in general, low drug tolerance or low drug adherence(6)". You have to critizise that this admittedly still predominant behavior makes no sense and must be changed in practice and in the guidelines. LAAC must be the first line treatment and OAC the treatment for those who cannot have LAAC or only have less than 2 years of life expectancy. LAAC has an initial risk of a severe problem of about 3-5% but then lacks the annual risk of a severe bleeding of about 2% germane to OAC. Equipoise in risk can be expected after 2 years. Thereafter, LAAC accumulates an ever growing advantage. Please change the respective statements in Introduction and Discussion.

When mentioning that 12 patients were excluded because they had no discharge visit, you have to specify what that means. So, I suggest to change the sentence to something like: "Twelve patients were excluded for reasons explained under Results."

In your new group of 62 same day discharge patients, you have 1 patient each with pericardial effusion witout tamponade and 1 with pericardial effusion with tamponade on the first day (their only day after the procedure and their day of discharge). Did you really discharge these patients the same day and, if yes, what happened to them?

You also had 1 patient with a TEE related complication in that group. What was it and was the patient really discharged the same day?

These patients are not in keeping with your statement: "Obviously a successful procedure without any SAE is a precondition for both time frames." which not only refers to the same day discharge patients but also to the next day discharge patients.

Reviewer #3: The authors have deeply revised the manuscript, addressing all the Reviewer's questions and Editorial requests.

7. PLOS authors have the option to publish the peer review history of their article (what does this mean?). If published, this will include your full peer review and any attached files.

Reviewer #1: **Yes: **Bernhard Meier

Reviewer #3: No

---

## [Author Response · Author response to Decision Letter 1]

23 May 2021

Dear reviewers,

thank you very much for your thoughtfull comments. We adressed them all, revised the manuscript and are eager to show the result. Below you will find the point to point response. 

Reviewer #1: The amendments are in part adequate. Yet, you need to dwell more on your new group of same day discharge patients which is the most interesting group of all. 

Moreover, you completely misread my initial comment. Your answer is the opposite of what I expected and intended to achieve. Please read my statement more carefully and amend your text accordingly: After 2 randomized trials showing a mortality benefit of left atrial appendage closure (LAAC) over oral anticoagulation (OAC) which materializes after 2 years and becomes more and more conspicuous with time, you cannot (cannot!) state that "However, percutaneous left atrial appendage occlusion (LAAO) has emerged as a favorable non pharmacological stroke prevention strategy in patients with AF who are unable to tolerate long-term anticoagulation use due to contraindications, high bleeding risk in general, low drug tolerance or low drug adherence(6)". You have to critizise that this admittedly still predominant behavior makes no sense and must be changed in practice and in the guidelines. LAAC must be the first line treatment and OAC the treatment for those who cannot have LAAC or only have less than 2 years of life expectancy. LAAC has an initial risk of a severe problem of about 3-5% but then lacks the annual risk of a severe bleeding of about 2% germane to OAC. Equipoise in risk can be expected after 2 years. Thereafter, LAAC accumulates an ever growing advantage. Please change the respective statements in Introduction and Discussion. 

• Reply: We thank the reviewer for his clear statement on the benefit of LAAO in patients with atrial fibrillation. In our introduction, we tried to fit in current guideline recommendations and underlined the growing evidence of further benefits of LAAO, which are not yet reflected in those recommendations. (p. 3 l- 60-70)

According to the guidelines a long-term anticoagulative strategy remains the first line therapy for stroke prevention with a Class I Level A recommendation, despite evident side effects and limited drug adherence.(5, 6) However, percutaneous left atrial appendage occlusion (LAAO) has emerged as a favorable non-pharmacological stroke prevention. After long term follow-up several randomized trials showed reductions in major bleeding, hemorrhagic stroke, and mortality as compared to warfarin (7). Despite positive clinical outcomes in trials of subjects tolerant of long-term anticoagulation, LAAO is currently only recommended for patients with AF who are unable to tolerate long-term anticoagulation use due to contraindications, high bleeding risk in general, low drug tolerance or low drug adherence(8) at a Class IIb Level B recommendation (6). 

When mentioning that 12 patients were excluded because they had no discharge visit, you have to specify what that means. So, I suggest to change the sentence to something like: "Twelve patients were excluded for reasons explained under Results."

• Reply: We thank the reviewer for the suggestion and have updated the Figure 1 legend accordingly.

In your new group of 62 same day discharge patients, you have 1 patient each with pericardial effusion without tamponade and 1 with pericardial effusion with tamponade on the first day (their only day after the procedure and their day of discharge). Did you really discharge these patients the same day and, if yes, what happened to them? 

• Reply: This is a very precise observation and we checked our files again. In reviewing the analysis the 1 Same Day Discharge patient with a pericardial effusion without tamponade was a transcription error into the manuscript. This value should instead be 0.0% and has been corrected in the revised manuscript. Hospital records were reviewed for the 1 Same Day Discharge patient with a pericardial effusion resulting in tamponade. By reviewing the hospital records it became apparent that the patient was discharged on day 5 post-LAAO as opposed to the same day (i.e., day 0) discharge provided by site-submitted data to the study database. Therefore, this patient‘s length of stay was updated to 5 days and the patient was recategorized from the Same Day Discharge to the Late Discharge group in the revised manuscript.

You also had 1 patient with a TEE related complication in that group. What was it and was the patient really discharged the same day? 

• Reply: Hospital records were reviewed for this adverse event. Shortly after the LAAO intervention bleeding from the nasopharynx / upper palate was observed. The source of bleeding was identified as a lesion in the upper palate, which stopped after a local injection of Suprarenin. An additional diffuse mucous membrane nasal bleed was controlled with insertion of a tamponade. Moreover the in-patient recovery was complicated by respiratory distress and a febrile infection requiring antibiotic therapy. These complications resulted in the subject actually having a 21 day in-hospital stay following LAAO, as opposed to the same day discharge indicated by site-submitted data to the study database. Therefore, this patient‘s length of stay was updated to 21 days and the patient recategorized from the Same Day Discharge to the Late Discharge group in the revised manuscript. 

All calculations were updated throughout the manuscript and figures updated accordingly.

These patients are not in keeping with your statement: "Obviously a successful procedure without any SAE is a precondition for both time frames." which not only refers to the same day discharge patients but also to the next day discharge patients. 

• Reply: We agree with the reviewer that using the term „precondition“ is too strong. We have reworded this text to reflect the extremely high degree of patients with successful procedures (same day= 100%; early discharge = 99.8%) and low degree of patients with procedure/device related SAEs prior to discharge (same day= 0%; early discharge= 1.0%). The text now reads: A successful procedure without any SAE often occurred for patients discharged within either timeframe and may be the primary factor for allowing same day or early discharge. 

Reviewer #3: The authors have deeply revised the manuscript, addressing all the Reviewer's questions and Editorial requests.

• Reply: We thank Reviewer #3 for their feedback leading to an improved manuscript through the revision process.

---

## [Decision Letter · Decision Letter 2]

23 Jul 2021

Length of stay following percutaneous left atrial appendage occlusion: data from the prospective, multicenter Amplatzer™ Amulet™ Occluder Observational Study

PONE-D-20-26960R2

Dear Dr. Zeus,

We’re pleased to inform you that your manuscript has been judged scientifically suitable for publication and will be formally accepted for publication once it meets all outstanding technical requirements.

Kind regards,

Giuseppe Andò, M.D., Ph.D.

Academic Editor

PLOS ONE

Additional Editor Comments (optional):

Reviewers' comments:

Reviewer's Responses to Questions

**Comments to the Author**

1. If the authors have adequately addressed your comments raised in a previous round of review and you feel that this manuscript is now acceptable for publication, you may indicate that here to bypass the “Comments to the Author” section, enter your conflict of interest statement in the “Confidential to Editor” section, and submit your "Accept" recommendation.

Reviewer #1: All comments have been addressed

2. Is the manuscript technically sound, and do the data support the conclusions?

Reviewer #1: Yes

3. Has the statistical analysis been performed appropriately and rigorously? 

Reviewer #1: Yes

4. Have the authors made all data underlying the findings in their manuscript fully available?

Reviewer #1: Yes

5. Is the manuscript presented in an intelligible fashion and written in standard English?

Reviewer #1: Yes

6. Review Comments to the Author

Reviewer #1: No further queries after the second revision. Thank you for the carefully done amendments. They improve the paper significantly.

7. PLOS authors have the option to publish the peer review history of their article (what does this mean?). If published, this will include your full peer review and any attached files.

Reviewer #1: No

---

## [Editor Report · Acceptance letter]

2 Aug 2021

PONE-D-20-26960R2 

Length of stay following percutaneous left atrial appendage occlusion: data from the prospective, multicenter Amplatzer Amulet Occluder Observational Study 

Dear Dr. Zeus:

I'm pleased to inform you that your manuscript has been deemed suitable for publication in PLOS ONE. Congratulations! Your manuscript is now with our production department. 

Kind regards, 

on behalf of

Dr. Giuseppe Andò 

Academic Editor

PLOS ONE